# Morphological and Phylogenetic Characterisations Reveal Four New Species in *Leptosphaeriaceae* (*Pleosporales*, *Dothideomycetes*)

**DOI:** 10.3390/jof9060612

**Published:** 2023-05-26

**Authors:** Ying Gao, Antonio Roberto Gomes de Farias, Hong-Bo Jiang, Samantha C. Karunarathna, Jian-Chu Xu, Saowaluck Tibpromma, Heng Gui

**Affiliations:** 1Center of Excellence in Fungal Research, Mae Fah Luang University, Chiang Rai 57100, Thailand; gaoying@mail.kib.ac.cn (Y.G.); antonio.gom@mfu.ac.th (A.R.G.d.F.); 2Center for Mountain Futures, Kunming Institute of Botany, Chinese Academy of Sciences, Honghe 654400, China; hongbo-j@hotmail.com (H.-B.J.); jxu@mail.kib.ac.cn (J.-C.X.); 3School of Science, Mae Fah Luang University, Chiang Rai 57100, Thailand; 4Department of Economic Plants and Biotechnology, Yunnan Key Laboratory for Wild Plant Resources, Kunming Institute of Botany, Chinese Academy of Sciences, Kunming 650201, China; 5National Institute of Fundamental Studies (NIFS), Kandy 20000, Sri Lanka; samanthakarunarathna@gmail.com; 6Center for Yunnan Plateau Biological Resources Protection and Utilization, College of Biological Resource and Food Engineering, Qujing Normal University, Qujing 655011, China

**Keywords:** grasslands, *Leptosphaeria*, multilocus phylogeny, *Paraleptosphaeria*, *Plenodomus*, taxonomy

## Abstract

*Leptosphaeriaceae* is a widely distributed fungal family with diverse lifestyles. The family includes several genera that can be distinguished by morphology and molecular phylogenetic analysis. During our investigation of saprobic fungi on grasslands in Yunnan Province, China, four fungal taxa belonging to *Leptosphaeriaceae* associated with grasses were collected. Morphological observations and phylogenetic analyses of the combined SSU, LSU, ITS, *tub*2, and *rpb*2 loci based on maximum likelihood and Bayesian inference were used to reveal the taxonomic placement of these fungal taxa. This study introduces four new taxa, viz. *Leptosphaeria yunnanensis*, *L. zhaotongensis*, *Paraleptosphaeria kunmingensis*, and *Plenodomus zhaotongensis*. Colour photo plates, full descriptions, and a phylogenetic tree to show the placement of the new taxa are provided.

## 1. Introduction

*Leptosphaeriaceae*, introduced by Barr [1], are widely distributed and exhibit diverse lifestyles, including fungicolous, epiphytic, parasitic, saprobic, and hemibiotrophic on herbaceous and woody plants [2,3,4,5,6]. For example, *Leptosphaeria* species are saprobes, plant pathogens, or hemibiotrophs on cultivated, wild herbaceous, and woody plants [7,8,9,10,11]. In the order *Pleosporales*, the family *Leptosphaeriaceae* contains economically significant plant pathogens [2]: i.e., Zhang et al. [8] reported *Leptosphaeria* species cause a serious disease of oilseed rape (*Brassica napus*, canola) in China.

The sexual morphs of *Leptosphaeriaceae* species are characterised by immersed to superficial and conical or globose ascomata, a thick peridium with scleroplectenchymatous or plectenchymatous cells, cylindrical to oblong asci, and hyaline to brown ascospores that are transversely septate [2,3,6,12,13,14,15]. In addition, asexual morphs are coelomycetous or hyphomycetous [13,16,17]. Furthermore, the most recently evolved *Leptosphaeria* species produce paler, longer, fusiform, and narrower ascospores with three or multi-septa, compared with the primitive dark brown ascospores [2,5].

*Leptosphaeria* and *Paraleptosphaeria* are morphologically similar, but phylogenetic analysis can clearly distinguish them [9,17,18,19,20]. The same applies to *Plenodomus*, and the other six genera, viz. *Alloleptosphaeria*, *Alternariaster*, *Neoleptosphaeria*, *Pseudoleptosphaeria*, *Sphaerellopsis*, and *Subplenodomus,* that were re-circumscribed based on molecular phylogeny [2]. Based on the multilocus phylogeny of the combined LSU, SSU, and ITS datasets, seven other genera, viz. *Angularia, Chaetoplea* [21], *Ochraceocephala* [22], *Praeclarispora* [23], *Heterosporicola*, *Querciphoma*, and *Sclerenchymomyces* [6], have been described in *Leptosphaeriaceae*. The Index Fungorum [24] listed 1682 *Leptosphaeria* species (accessed on 29 March 2023) epithets, and many of them have been synonymised under other genera. The members of *Paraleptosphaeria* have been reported as saprobic, fungicolous, or pathogenic [2,4,20], and nine species epithets are listed in the Index Fungorum [24] (accessed on 29 March 2023). In addition, *Plenodomus* has often been reported as saprobic or pathogenic on *Asteraceae*, *Fabaceae*, *Lamiaceae*, *Liliaceae,* and *Poaceae* [2,6,13,25,26] and 101 records are listed in the Index Fungorum [24] (accessed on 29 March 2023).

Grasslands comprise a biome subjected to alternating drought, where grass and grass-like species dominate, and arboreous trees are uncommon [27]. In the grassland biome, several living organisms, such as herbivorous mammals, insects, and fungi (pathogenic, saprobic, and symbiotic), play essential roles in maintaining biomass and biodiversity [28]. Regarding fungi, a checklist of *Ascomycetes* on grasses was provided by Karunarathna et al. [29], which lists 3,165 fungal species. Studies of fungi on grasses include those of Thambugala et al. [30], Goonasekara et al. [31], and Brahmanage et al. [32].

This study describes four new *Leptosphaeriaceae* species that were collected from herbaceous plants in Kunming and Zhaotong, Yunnan Province, China. Phylogenetic analyses results based on SSU, LSU, ITS, *tub*2, and *rpb*2 loci, colour photo plates, complete descriptions of the four new species, and a summary of the morphological characteristics of *Leptosphaeria*, *Paraleptosphaeria*, and *Plenodomus* are provided.

## 2. Materials and Methods

### 2.1. Sample Collection, Isolation, and Identification

Herbaceous plants with fungal fruiting bodies were collected from Kunming and Zhaotong in Yunnan Province, China, stored in plastic bags, and returned to the mycology laboratory at the Kunming Institute of Botany. The samples were examined under an Olympus SZ-61 dissecting microscope (Tokyo, Japan). Fungal fruiting bodies were manually sectioned and mounted in double distilled water (ddH_2_O). Micro-morphological characteristics were captured using an OLYMPUS SZ2-ILST compound microscope connected to an Industrial Digital Camera 16NP USB3.0 (Panasonic, Osaka, Japan) microscope imaging system. Photo plates were processed using Adobe Photoshop CS6 Extended version 13.0.1 (Adobe Systems, San Jose, CA, USA). As described by Senanayake et al. [33], cultures were obtained via single-spore isolation and incubated under normal light at room temperature (25 °C). Germinating ascospores or conidia were observed under a stereo microscope and transferred to new potato dextrose agar (PDA) plates. Herbarium specimens were deposited in the herbarium at the Kunming Institute of Botany, Chinese Academy of Sciences (HKAS), and the Herbarium Mycologicum Academiae Sinicae, Beijing, China (HMAS), and living cultures were deposited in the China General Microbiological Culture Collection Center (CGMCC). The Index Fungorum [24] and Faces of Fungi (FoF) [34] numbers were registered for the new species.

### 2.2. DNA Extraction, PCR Amplification, and DNA Sequencing

Genomic DNA was extracted from fresh mycelia grown on PDA at 28 °C for two weeks using the Biospin Fungus Genomic DNA Extraction Ki-BSC14S1 (BioFlux^®^, Hangzhou, China) according to the manufacturer’s protocol. The E.Z.N.A. Forensic DNA Kit-D3591 (Omega Biotek, Inc., Norcross, Georgia) was used to extract DNA directly from fruiting bodies. Polymerase chain reactions (PCRs) were carried out for five genetic markers: internal transcribed spacer region (ITS) [35], partial 28S large subunit nuclear ribosomal DNA (LSU) [36], partial small subunit ribosomal RNA (SSU) [35], β-tubulin (*tub*2) [37], and partial RNA polymerase second largest subunit (*rpb*2) [38]. The primers and amplification conditions used are listed in Table 1. The total volume of PCR mixtures for amplification was 25 μL, containing 8.5 μL ddH_2_O, 12.5 μL 2xF8FastLong PCR MasterMix (Beijing Aidlab Biotechnologies Co. Ltd., Beijing, China), 2 μL of the DNA template, and 1 μL each of reverse and forward primer (10 pM). The PCR products were sequenced by Shanghai Sangon Biological Engineering Technology and Service Co., Ltd., Shanghai, China.

### 2.3. Phylogenetic Analyses

The newly obtained sequences (SSU, LSU, ITS, *tub*2, and *rpb*2) were subjected to BLASTn searches against the GenBank database (https://blast.ncbi.nlm.nih.gov/Blast.cgi) (accessed on 29 March 2023) to identify closely related taxa. Reference isolates and accessions were obtained from recent studies [6,17,23,39,40,41] and downloaded from GenBank. Single-locus sequence datasets were aligned using MAFFT v. 7.505 [42], trimmed using TrimAl v. 1.3 [43] via the web server Phylemon2 (http://phylemon.bioinfo.cipf.es/utilities.html) (accessed on 29 March 2023), and concatenated using BioEdit v. 7.0.5.2 [44]. Phylogenetic reconstructions of individual and combined datasets were performed using maximum likelihood (ML) and Bayesian inference (BI) analyses.

Maximum likelihood trees were inferred using RAxML-HPC2 in the XSEDE v. 8.2.12 [45] in CIPRES Science Gateway v. 3.3 online platform [46] under the GTRGAMMA nucleotide substitution model with 1000 bootstrap replicates. Bayesian inference analysis was conducted using MrBayes on XSEDE v. 3.2.7a [47], under the substitution model GTR + I + G for all loci, estimated by MrModeltest v. 2.3 [48] using PAUP v. 4.0b10 [49]. Six simultaneous Markov chains were run for 10,000,000 generations, with trees sampled every 1000th generation. The run was configured to stop when the standard deviation of split frequencies dropped below 0.01, and the first 25% of the trees were discarded as burn-in.

Tree topologies were visualised and exported using FigTree v. 1.4.0 [50]. The phylogram was edited and annotated using Microsoft Office PowerPoint 2016 (Microsoft Inc., Redmond, WA, USA) and Adobe Photoshop CS6 Extended version 13.0.1 (Adobe Systems, San Jose, CA, USA). Finally, the newly generated sequences were deposited in the GenBank database (Table 2).

The decision as to whether the new species should be introduced followed the polyphasic guidelines of Chethana et al. [51] and Maharachchikumbura et al. [52].

## 3. Results

### 3.1. Phylogenetic Analyses

The combined sequence data of SSU, LSU, ITS, *tub*2, and *rpb*2 consisted of 177 strains of *Leptosphaeriaceae*, plus *Didymella exigua* (CBS 183.55) and *D. maydis* (CBS 588.69) as outgroup taxa (Figure 1). After trimming, the dataset consisted of 3981 sites, including gaps (SSU = 1–894 bp, LSU = 895–2224 bp, ITS = 2225–2780 bp, *tub*2 = 2781–3108 bp, and *rpb*2 = 3109–3981 bp). The phylogenetic tree topologies of the single and combined matrices were similar. The phylogenetic topologies obtained using the ML and BI methods also shared the same topology. RAxML analysis of the combined dataset yielded a best-scoring tree with a final ML optimization likelihood value of −31,560.945159. The matrix had 1315 distinct alignment patterns, with 45.47% undetermined characters or gaps. Estimated base frequencies were as follows: A = 0.243300, C = 0.230842, G = 0.272923, T = 0.252935; substitution rates were as follows: AC = 1.583725, AG = 4.540133, AT = 1.817809, CG = 0.949937, CT = 7.364395, GT = 1.00, alpha: 0.147156.

Our phylogenetic analysis of *Leptosphaeriaceae* is analogous to the analysis by Xu et al. [6]. Our isolate, *Leptosphaeria zhaotongensis* (HKAS 124664, HMAS 352282), is closely related to *L. cichorii* (MFLUCC 14-1063) with 96% ML and 1.00 BYPP bootstrap support (Figure 1). *Leptosphaeria yunnanensis* (CGMCC 3.23748, CGMCC 3.23749, HKAS 124671) formed a well-separated lineage from other *Leptosphaeria* species with 67% ML and 0.91 BYPP bootstrap support (Figure 1). *Plenodomus zhaotongensis* (CGMCC 3.23746, CGMCC 3.23747) is closely related to the *Pl. agnitus* strains with 91% ML and 1.00 BYPP bootstrap support (Figure 1). *Paraleptosphaeria kunmingensis* (KUNCC 23-12732, KUNCC 23-12731) is closely related to *Pa. macrospora* (CBS 114198) with 88% ML and 1.00 BYPP bootstrap support (Figure 1).

### 3.2. Taxonomy

#### 3.2.1. ***Leptosphaeria yunnanensis*** Y. Gao and H. Gui, sp. nov.

Index Fungorum number: IF 556122; Faces of Fungi number: FoF 12905; Figure 2.

**Etymology:** The specific epithet “yunnanensis” refers to Yunnan Province, where the holotype was collected.

**Holotype:** HKAS 124670

*Saprobic* on a decaying stalk of herbaceous plant. **Sexual morph:** Undetermined. **Asexual morph:**
*Conidiomata* 280–515 μm diam × 170–370 μm high ($\overset{-}{x}$
*=* 385 × 253 μm, *n* = 20), in small groups or scattered, solitary, erumpent to superficial, subglobose to globose, smooth-walled, easily removed from the host substrate, black, coriaceous, without ostiolate. *Conidiomatal wall* (132–)144–189(–202) μm thick, ($\overset{-}{x}$
*=* 166 μm, *n* = 30), thick, almost fills the entire conidiomata, each cell-layer (9–)10.6–15.6(–18) μm wide, ($\overset{-}{x}$
*=* 13 μm, *n* = 40), composed of flattened cells of *textura angularis*, highly pigmented on the outside and hyaline on the inside, with conidiogenous cells. *Conidiogenous cells* (2.8–)3.6–5.2(–6.4) μm long × (3.8–)4.4–5.8(–6.7) μm wide ($\overset{-}{x}$
*=* 4.4 × 5 μm, *n* = 40), ampulliform or globose to subglobose, smooth-walled, hyaline. *Conidia* (2.5–)3.3–5.6(–6.6) μm long × (1–)1.2–1.5(–1.8) μm wide ($\overset{-}{x}$
*=* 4.5 × 1.4 μm, *n* = 30), ellipsoidal to sub-cylindrical with obtuse ends, hyaline, guttulate, aseptate, smooth-walled.

**Culture characteristics:** Conidia germinated on PDA within 20 h. Colonies on PDA reaching 20 mm at four weeks at room temperature (25–27 °C), hairy or cottony, raised, white to grey, mycelium superficial, dark brown at the margin, white to light grey at the centre from above, grey in the centre, gradually black towards the edges from below.

**Material examined:** China, Yunnan Province, Zhaotong City, Daguan County, grassland (27°44′23″ N, 103°47′59″ E), on a decaying stalk of herbaceous plant, 21 August 2021, Ying Gao, ZG25A (HKAS 124670, holotype), ex-type, CGMCC 3.23748; ibid., ZG25C (HKAS 127125, paratype), ex-paratype CGMCC 3.23749; ibid., ZG25 (HKAS 124671, paratype).

**Notes:** Based on multilocus phylogenetic analyses, our isolates of *L. yunnanensis* (CGMCC 3.23748, CGMCC 3.23749, and HKAS 124671) showed a well-separated lineage within *Leptosphaeria* with moderate statistical support (67% ML, 0.91PP (Figure 1)). It clustered between *L. urticae* (MFLU 18-0591) and *L. pedicularis* (CBS 390.80) (Figure 1). The pairwise nucleotide comparison showed that our strain (CGMCC 3.23748) differs from *L. urticae* (MFLU 18-0591) in 44/487 bp of ITS (9.03%, with 5 gaps) and *L. pedicularis* (CBS 390.80) in 37/513 bp of ITS (7.23%, with 6 gaps) and 24/334 bp of *tub*2 (7.19%, with 2 gaps). Significant morphological differences were observed when compared with the literature for the genus (for example, [2,5,13,41,53]). *Leptosphaeria yunnanensis* differs from related species by having a unique conidiomatal wall that occupies almost the entire interior of the conidiomata and sporulation indistinctly in the centre. In addition, *Leptosphaeria yunnanensis* was reported as an asexual form, and although it is similar to *L. cichorii* in conidiomata and conidiogenous cells, the sizes are different (conidiomata 280–515 × 170–370 μm vs. 189–200 × 196–220 μm) (conidiogenous cells 2.8–6.5 × 3.5–6.5 μm vs. 2–5 × 2–4 μm) [2]. Furthermore, these two species formed a well-separated lineage (Figure 1). Therefore, based on the polyphasic approach recommended for species-boundary delimitation [51,52], we introduce *L. yunnanensis* as a novel taxon.

#### 3.2.2. ***Leptosphaeria zhaotongensis*** Y. Gao and H. Gui, sp. nov.

Index Fungorum number: IF 556123; Faces of Fungi number: FoF 12904; Figure 3.

**Etymology:** The specific epithet “zhaotongensis” refers to Zhaotong City, where the holotype was collected.

**Holotype:** HKAS 124664

*Saprobic* on a decaying stalk of a herb. **Sexual morph**: *Ascomata* 380–550 μm diam × 185–300 μm high ($\overset{-}{x}$
*=* 461 × 234 μm, *n* = 15), scattered or in small groups, solitary, erumpent to superficial, globose to ampulliform, smooth-walled, easily removed from the host substrate, with a flattened base, black, coriaceous, uni-loculate, glabrous, ostiolate. *Ostiole* apex conical and with papilla. *Peridium* (35–)66–116.5(–127) μm wide, ($\overset{-}{x}$
*=* 91 μm, *n* = 40), thick-walled, composed of 4–8 layers of flattened, light brown to dark brown cells of *textura angularis*. *Hamathecium* (1.5–)2–2.6(–3) μm wide, ($\overset{-}{x}$
*=* 2.3 μm, *n* = 50), straight, septate, hyaline, branched, cellular pseudoparaphyses, partially embedded in a gelatinous matrix. *Asci* (84–)92–113(–118) × (7.7–)8.2–10(–11) μm ($\overset{-}{x}$
*=* 102 × 9 μm, *n* = 30), eight-spored, arising from the base, cylindrical to cylindric-clavate, bitunicate, fissitunicate, apically rounded, short pedicellate with foot-like pedicel, with ocular chamber, hyaline. *Ascospores* (16.2–)17.6–20(–21.3) × (4.5–)4.8–5.7(–6.5) μm ($\overset{-}{x}$
*=* 18.7 × 5.2 μm, *n* = 40), uniseriate, fusiform, partially overlapping, narrow to acute at both ends, guttulate, initially hyaline with one septum, becoming yellowish to brown, 3-septate at maturity, broader cells above central septum, often slightly constricted at septum, sometimes rough-walled, without mucilaginous sheath. **Asexual morph:** Undetermined.

**Material examined:** China, Yunnan Province, Zhaotong City, Daguan County, grassland (27°44′23″ N, 103°47′59″ E), on a decaying stalk of herbaceous plant, 21 August 2021, Ying Gao, GG (HKAS 124664, holotype); ibid., (HMAS 352282, paratype).

**Notes:** Based on our phylogenetic analysis of the combined SSU, LSU, ITS, *tub2*, and *rpb*2 sequence data, our novel species *L. zhaotongensis* (HKAS 124664, HMAS 352282) is closely related to *L. cichorii* (MFLUCC 14-1063) with 96% ML and 1.00 PP statistical support (Figure 1). *Leptosphaeria zhaotongensis* differs from *L. cichorii* in its larger ascomata (384–551 × 186–292 μm vs. 206–240 × 251–363 μm) and ascospores (16–21 × 5–7 μm, hyaline, yellowish to brown, guttulate vs. 11–20 × 3–6 μm, reddish to yellowish brown, without guttulate). The similarity of the ITS sequence data of *L. cichorii* (MFLUCC 14-1063) was 62/521 bp (11.9%, with 6 gaps) compared to *L. cichorii* (MFLUCC 14-1063). Therefore, based on the polyphasic approach recommended for species boundaries delimitation [51,52], we introduce *L. zhaotongensis* as a novel taxon.

#### 3.2.3. ***Paraleptosphaeria kunmingensis*** Y. Gao and H. Gui, sp. nov.

Index Fungorum number: IF 556124; Faces of Fungi number: FoF 12902; Figure 4.

**Etymology:** The specific epithet “kunmingensis” refers to Kunming City, where the holotype was collected.

**Holotype:** HKAS 124662

*Saprobic* on a decaying stalk of herbaceous plant. **Sexual morph:** *Ascomata* 215–300 μm × 145–220 μm ($\overset{-}{x}$
*=* 253 × 176 μm, *n* = 15), scattered, gregarious, immersed in the epidermis of the host, globose or subglobose and flat-globose, dark brown to black, uni-loculate, glabrous, shiny, papillate, with ostiole. *Peridium* (21–)24–34(–43) μm wide, ($\overset{-}{x}$
*=* 29 μm, *n* = 35), composed of 2–4 layers of flattened, light brown to dark brown cells of *textura angularis*. *Hamathecium* (1.6–)2–3(–3.7) μm wide, ($\overset{-}{x}$
*=* 2.5 μm, *n* = 40), straight, septate, hyaline, unbranched, cellular pseudoparaphyses, embedded in a gelatinous matrix. *Asci* (70–)73–92(–104) × (12–)13–15.7(–16.4) μm ($\overset{-}{x}$
*=* 83 × 14 μm, *n* = 25), eight-spored, arising from base, fissitunicate, bitunicate, cylindrical to cylindric-clavate, short pedicellate with club-like pedicel, thick-walled at the apex, hyaline, with ocular chamber. *Ascospores* (33.5–)37.6–47(–50.5) × (5–)5.2–6.2(–7.2) μm ($\overset{-}{x}$
*=* 42.3 × 5.7 μm, *n* = 30), overlapping, 2–3-seriate, hyaline, guttulate, lunate to long fusiform or inequilateral, straight or slightly curved, with 1–3 transverse septa, often slightly constricted at septum, swollen at the second cell, rounded to slightly pointed at both ends, guttulate, without a mucilaginous sheath. **Asexual morph:** Undetermined.

**Culture characteristics:** Ascospores germinated on PDA within 20 h, and a germ tube was initially produced from the middle. Colonies on PDA reaching 15 mm at two weeks at room temperature, circular, slightly raised, curled, floccose, pale yellow from above, dark brown in the centre, gradually pale yellow towards the edges from below, grows towards the filamentous edge.

**Material examined:** China, Yunnan Province, Kunming City, (25°8′19″ N, 102°44′25″ E), on a decaying stalk of herbaceous plant, 20 June 2021, Ying Gao, CCSG18A (HKAS 124662, holotype), ex-type KUNCC 23-12732. ibid., CCSG18 (HKAS 127126, paratype), ex-paratype KUNCC 23-12731.

**Notes:** *Paraleptosphaeria kunmingensis* is introduced as a new species based on its distinct morphology and phylogenetic analysis of combined SSU, LSU, ITS, *tub*2, and *rpb*2 datasets. *Paraleptosphaeria kunmingensis* is closely related to *Pa. macrospora* (CBS 114198) with 88% ML and 1.00 BYPP bootstrap support (Figure 1). The species differs from *Pa. macrospora* (Basionym: *Metasphaeria macrospora*) in its smaller asci (83 × 14 μm vs. 105 × 18 μm), smaller ascospores (42.3 × 5.7 μm vs. 44 × 8 μm), and the number of septa of ascospores (1–3 septa vs. three septa) [13,54]. In addition, the ITS pairwise nucleotide comparison of these species showed 18/523 bp differences (3.44%, without gaps). Our isolate differs from *Pa. nitschkei* in 7.51% (KT389833) and *Pa. dryadis* (GU371733) in 10.05% in the *tub*2 and *rpb*2 regions, respectively. Therefore, based on the polyphasic approach recommended for species boundaries delimitation [51,52], we introduce *Pa. kunmingensis* as a novel taxon.

#### 3.2.4. ***Plenodomus zhaotongensis*** Y. Gao and H. Gui, sp. nov.

Index Fungorum number: IF 556124; Faces of Fungi number: FoF 12903; Figure 5.

**Etymology:** The specific epithet “zhaotongensis” refers to Zhaotong City, where the holotype was collected.

**Holotype:** HKAS 124668

*Saprobic* on a decaying stalk of herbaceous plant. **Sexual morph**: *Ascomata* 200–300 μm × 210–320 μm ($\overset{-}{x}$
*=* 240 × 298 μm, *n* = 15), scattered, most are gregarious, raised, superficial with base seated in the substrate, globose to subglobose or irregular, apically conical, dark brown to black, uni-loculate, glabrous, coriaceous, ostiolate, papillate ostiole, wider and flattened at the base, connected by very thin stromatal tissue at the base. *Peridium* (22–)29–54(–79) μm wide, ($\overset{-}{x}$
*=* 42 μm, *n* = 50), composed of two types of scleroplectenchymatous cells layers, thick-walled of unequal thickness, thickened at base, thinner toward sides and apex, inner layers composed of hyaline to pale brown cells of *textura angularis* to *textura globulosa*, outer layer of amorphous black cells. *Hamathecium* (1.4–)2–3.7(–6) μm wide, ($\overset{-}{x}$
*=* 3 μm, *n* = 45), septate, hyaline, unbranched, broad at base, tapering upwards, pseudoparaphyses. *Asci* (80–)97–120(–132) × (10–)11–13(–14) μm ($\overset{-}{x}$
*=* 109 × 12 μm, *n* = 20), eight-spored, arising from base, fissitunicate, bitunicate, cylindric-clavate, initially hyaline, short pedicellate with foot-like pedicel, with ocular chamber, thick-walled at the apex. *Ascospores* (34–)35–39(–42) × (3.5–)4–5(–6) μm ($\overset{-}{x}$
*=* 37 × 4.5 μm, *n* = 30), overlapping, 2–3-seriate, initially hyaline, becoming pale yellowish at maturity, guttulate, lunate to long fusiform, straight or slightly curved, with six transverse septa at maturity, often constricted at medium septum, widest at the middle, rounded or slightly pointed at both ends, without a mucilaginous sheath. **Asexual morph**: Undetermined.

**Culture characteristics:** Ascospores germinated on PDA within 20 h, and germ tube initially produced from both ends of the ascospores. Colonies on PDA reaching 20 mm at four weeks at room temperature, irregular, flat, centre is slightly raised, panniform, mycelium grows on the surface of PDA, white from above, brown in the centre gradually pale yellow towards the edges from below. Asexual spores and sexual spores were not formed on PDA within 60 days.

**Material examined:** China, Yunnan Province, Zhaotong City, Daguan County, grassland (27°44′23″ N, 103°47′59″ E), on a decaying stalk of herbaceous plant, 21 August 2021, Ying Gao, ZG17A (HKAS 124668, holotype), ex-type, CGMCC 3.23746. ibid., ZG17 (HKAS 127124, paratype), ex-paratype, CGMCC 3.23747.

**Notes:** *Plenodomus zhaotongensis* is introduced as a new species based on its distinct morphology and phylogenetic analysis of combined SSU, LSU, ITS, *tub2*, and *rpb*2 sequence data. *Plenodomus zhaotongensis* is closely related to *Pl. agnitus* strains with 91% ML and 1.00 BYPP statistical support (Figure 1). A pairwise nucleotide comparison showed that *Pl. zhaotongensis* differs from *Pl. agnitus* (CBS 121.89) in 12/529 bp of ITS (2.27%, without gaps), 12/341 bp of *tub*2 (3.52%, without gaps), and 18/766 bp of *rpb*2 (2.35%, without gaps). *Plenodomus zhaotongensis* differs from *Pl. agnitus* (sexual morph: *Leptosphaeria agnita* (Desm.) Ces. & De Not., Comm. Soc. Crittog. Ital. 1: 236. 1863.) in its larger ascospores (34–42 × 3.5–6 μm, lunate to long fusiform vs. 31–35 × 4–5 μm, narrowly subcylindrical) [13,55]. Therefore, based on the guidelines for new species boundaries delimitation [51,52], we introduce *Pl. zhaotongensis* as a novel taxon.

## 4. Discussion

In this study, we introduce four new species, viz. *L. yunnanensis*, *L. zhaotongensis*, *Pa. kunmingensis*, and *Pl. zhaotongensis,* associated with grasses from Zhaotong and Kunming in Yunnan Province, southwestern China, based on polyphasic approaches [56,57] through multilocus analyses of five gene loci (SSU, LSU, ITS, *tub2*, and *rpb*2) and their morphological characteristics. Although *Leptosphaeria* is a speciose genus with 1682 species epithets, many are likely to belong to other genera [2] and need recollecting and sequencing. However, in speciose genera, many novel taxa can still be found [58], as in this study.

*Paraleptosphaeria* and *Leptosphaeria* have similar morphologies. In our study, *Pa. kunmingensis* fits within the generic concept of *Paraleptosphaeria* and is phylogenetically closely related to *Pa. macrospora* (Figure 1), whereas *L. yunnanensis* and *L. zhaotongensis* clustered distinctly in *Leptosphaeria* (Figure 1). All species differ in morphology (Figure 2, Figure 3 and Figure 5, Appendix A). Using molecular data, Piątek et al. [20] also showed that *Paraleptosphaeria* and *Leptosphaeria* are phylogenetically distinct. We also described *Pl. zhaotongensis* based on the morphological characteristics (Figure 5) and molecular phylogeny (Figure 1). Nevertheless, in *Plenodomus*, it is challenging to have well-resolved species delimitation because many species lack molecular data and detailed morphological descriptions [5,21]. Therefore, in future studies, precise morphological characteristics, molecular data, and phylogenetic analyses should be provided for all newly introduced and existing *Plenodomus* species.

Except for *L. maculans*, leptosphaeria-like taxa are diverse and widespread. However, they are mostly found in temperate regions (Appendix A, [59,60]), with 30 species reported on grasses [29]. In addition, *Leptosphaeria* species seem not to be host-specific, as they have been discovered on various plant families (i.e., *Adoxaceae*, *Apiaceae*, *Asteraceae*, *Euphorbiaceae*, *Fabaceae*, *Gentianaceae*, *Juglandaceae*, *Lamiaceae*, *Orobanchaceae*, *Plantaginaceae*, *Rhamnaceae*, and *Urticaceae*) [5]. Similarly, *Plenodomus* is widely distributed worldwide, mainly in temperate countries such as China, Greece, France, Japan, the Netherlands, and Spain (Appendix A, [6,61]), with four species, viz. *Pl. acutus*, *Pl. changchunensis*, *Pl. enteroleucus*, and *Pl. sorghi,* are associated with grasses [6,13,62,63]. This indicates that investigations of new host plants, especially those inhabiting decomposing litter [64,65], and unstudied environments will result in undescribed taxa in this and other families, contributing to the descriptive fungal curve [66,67].

The four described species in this study were collected from climate-contrasting grasslands in Yunnan Province. In Zhaotong, subtropical and warm temperate zones coexist, with an annual average temperature of 12.6 °C [68,69], while Kunming has distinct wet and dry seasons [70,71]. In terms of fungi, many new fungal species have been reported in Yunnan Province in the last two decades. More than 1300 new fungal species have been described, accounting for nearly 25% of the total fungal species described in China [72]. This scenario is consistent with Hyde et al. [66,67], who stated that continued exploration of new environments would result in undescribed taxa.

In Yunnan’s grasslands, many grass species are the primary source of carbohydrates and feed for livestock, and fungi play a pivotal role in maintaining and shaping grass communities. Each grass-associated fungal community is responsible for specific ecological properties of the environment [29]. Our study fills some gaps in the research on *Leptosphaeriaceae* species in grasslands by providing detailed information on four new species from China and insights into the number of grass-associated *Leptosphaeria*. In addition, fungi have never been reported in Zhaotong City; thus, future studies are needed to reveal the actual fungal diversity, especially those associated with grasses.

In addition, we compared our *Leptosphaeria zhaotongensis* with known species that have no molecular data in Shoemaker [73]. *Leptosphaeria zhaotongensis* is similar to *L. galii, L. raphani*, *L. russellii*, *L. stellariae,* and *L. byssincola* in ascomata, asci and ascospores. However, they differ from *Leptosphaeria galii* in ascospores (guttulate vs. without guttulate) and asci (84–118 × 7.7–11 μm, vs. 45–65 × 6–8 μm); *Leptosphaeria zhaotongensis* differs from *Leptosphaeria raphani* in ascospores (guttulate vs. without guttulate), asci (84–118 μm, vs. 60–80 μm), and ascomata (380–550 × 185–300 μm, vs. 200–280 × 200–280 μm); *Leptosphaeria zhaotongensis* differs from *Leptosphaeria russellii* in asci (84–118 μm, vs. 70–85 μm) and ascomata (380–550 × 185–300 μm, vs. 200–250 × 100–150 μm); *Leptosphaeria zhaotongensis* differs from *Leptosphaeria stellariae* in ascospores (16–21 μm × 4.5–6.5 μm, guttulate, rough, vs. 26–30 × 6–7 μm, without guttulate, smooth), asci (84–118 μm, vs. 80–180 μm), and ascomata (380–550 × 185–300 μm, vs. 140–190 × 90–110 μm); and *Leptosphaeria zhaotongensis* differs from *Leptosphaeria byssincola* in asci (84–118 × 8–11 μm, vs. 80–90 × 11–13 μm). There are also obvious differences in morphology between *Leptosphaeria zhaotongensis* and unsequenced *Leptosphaeria* species.

## Figures and Tables

**Figure 1 jof-09-00612-f001:**
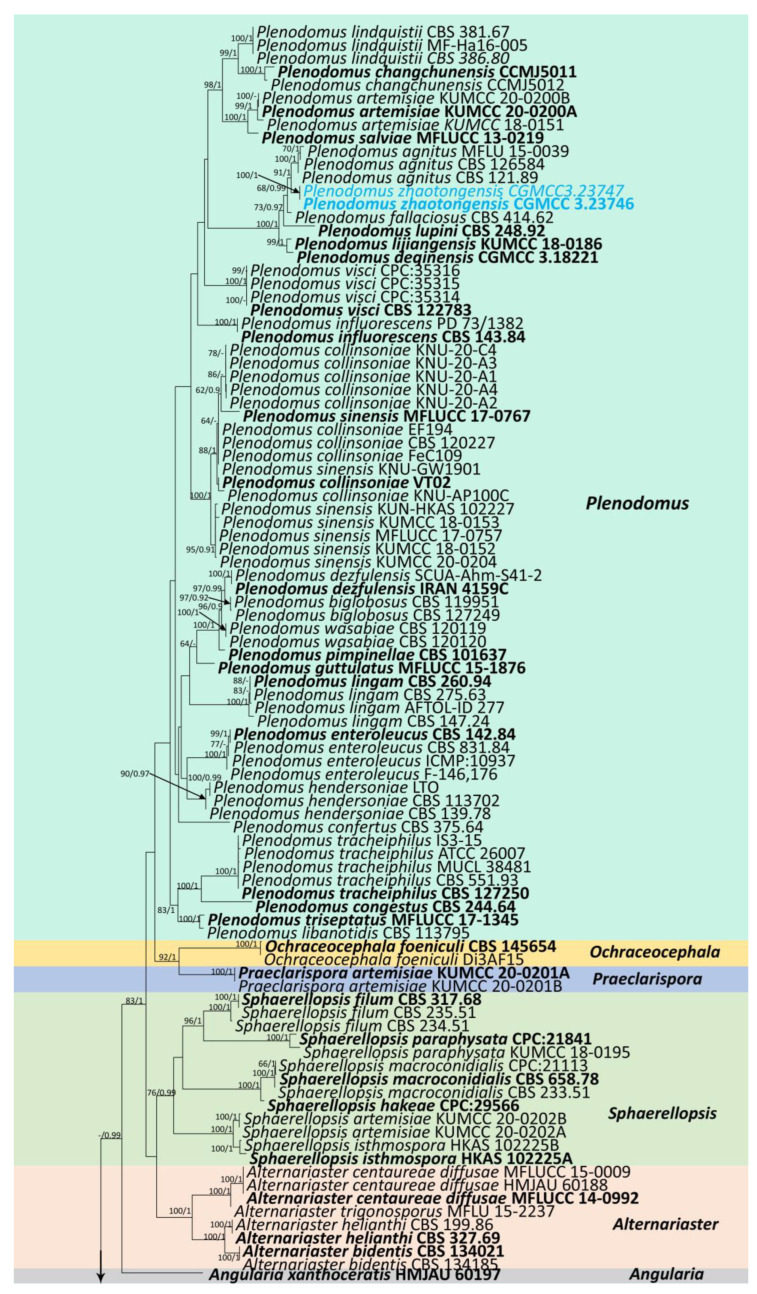
Phylogenetic tree obtained from combined SSU, LSU, ITS, *tub*2, and *rpb*2 sequence data. Numerical values at the nodes indicate bootstrap support, and maximum likelihood bootstrap support ≥65% and Bayesian posterior probabilities ≥0.90 are displayed at the node. Ex-type strains are in bold, and the newly generated sequences are shown in red.

**Figure 2 jof-09-00612-f002:**
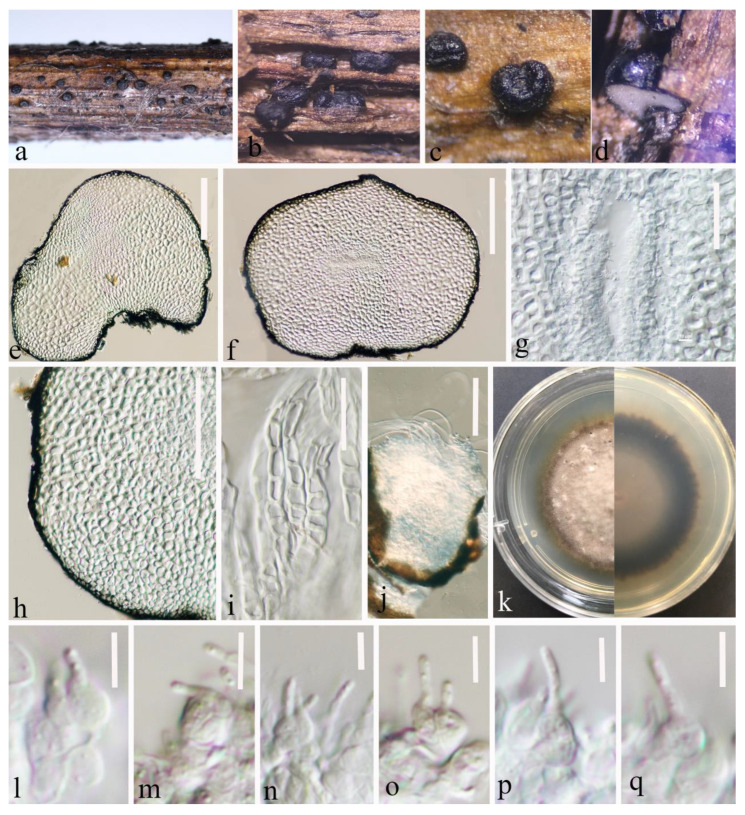
*Leptosphaeria yunnanensis* (HKAS 124670 holotype). (**a**–**d**) Black conidiomata on the host surface. (**e**,**f**) Vertical sections of conidiomata. (**g**) Conidia location in conidiomata. (**h**,**i**) Peridium. (**j**) Germinating conidiomata. (**k**) Front and reverse of colony on PDA. (**l**–**q**) Conidiogenous cells and developing conidia in conidiomata. Scale bars: (**e**,**f**) = 150 μm. (**g**) = 50 μm. (**h**) = 100 μm. (**i**) = 50 μm. (**j**) = 100 μm. (**l**–**q**) = 5 μm.

**Figure 3 jof-09-00612-f003:**
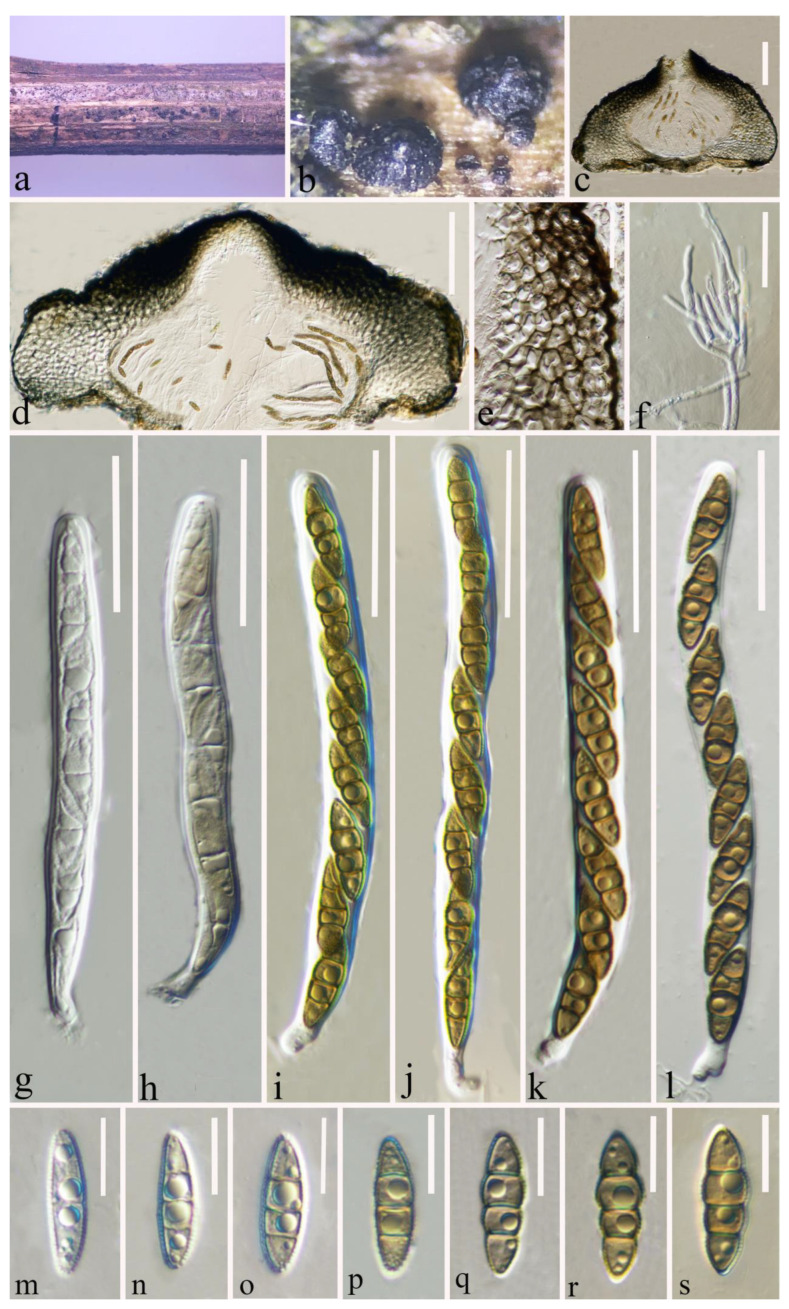
*Leptosphaeria zhaotongensis* (HKAS 124664 holotype). (**a**,**b**) Ascomata on decaying stalk of herbaceous plant. (**c**,**d**) Vertical section of the ascoma. (**e**) Peridium. (**f**) Hamathecium. (**g**–**l**) Asci. (**m**–**s**) Ascospores. Scale bars: (**c**–**d**) = 100 μm. (**e**) = 30 μm. (**f**) = 20 μm. (**g**–**l**) = 30 μm. (**m**–**s**) = 10 μm.

**Figure 4 jof-09-00612-f004:**
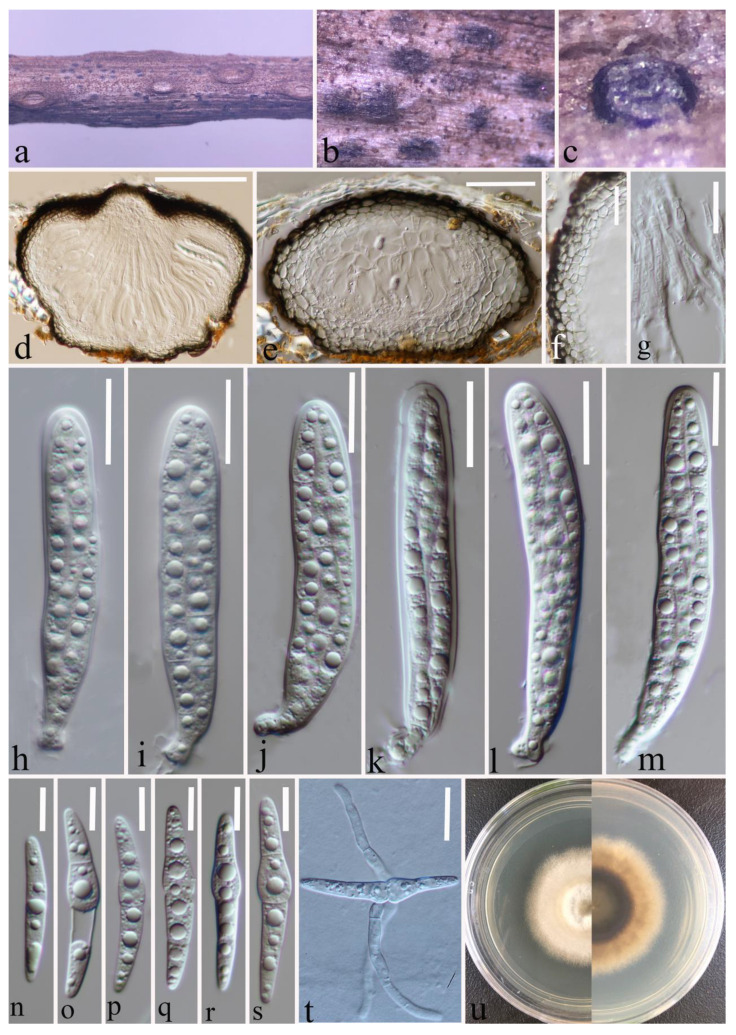
*Paraleptosphaeria kunmingensis* (HKAS 124662 holotype). (**a**–**c**) Ascomata on decaying stalk of herbaceous plant. (**d**,**e**) Vertical section of the ascoma. (**f**) Peridium. (**g**) Hamathecium. (**h**–**m**) Asci. (**n**–**s**) Ascospores. (**t**) Germinating ascospore. (**u**) Front and reverse colony on PDA. Scale bars: (**d**) = 100 μm, (**e**) = 50 μm, (**f**) = 30 μm, (**g**–**m**) = 10 μm, (**n**–**s**) = 10 μm, (**t**) = 20 μm.

**Figure 5 jof-09-00612-f005:**
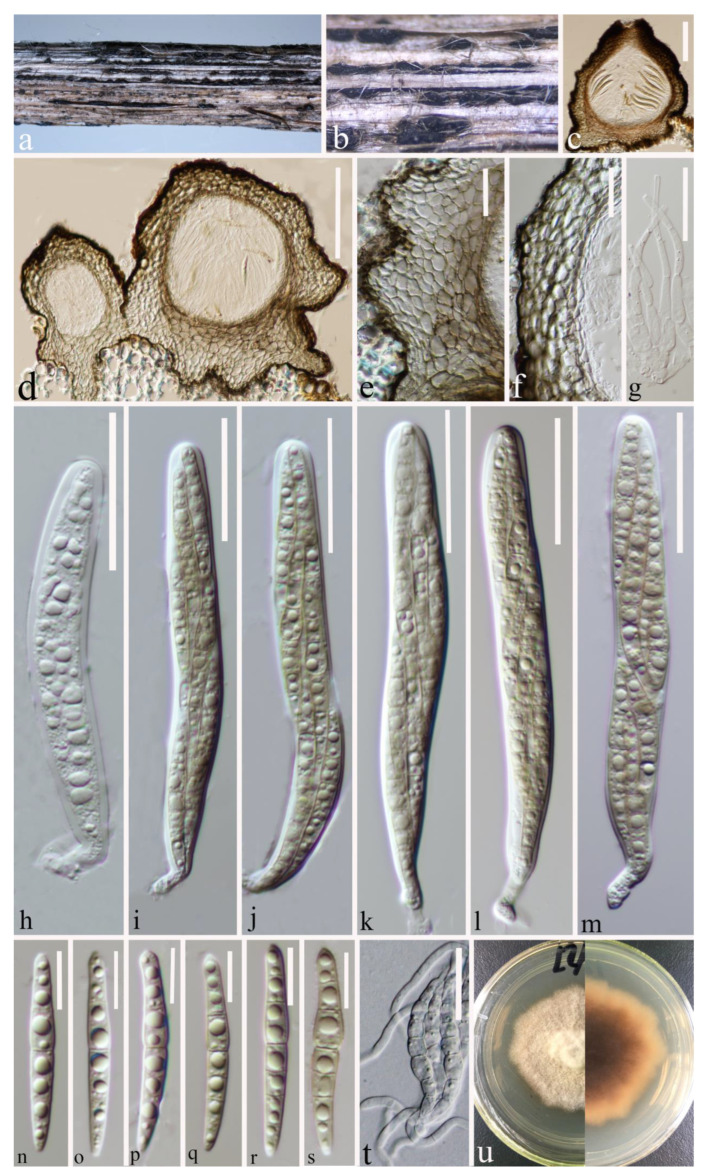
*Plenodomus zhaotongensis* (HKAS 124668 holotype). (**a**,**b**) Ascomata on decaying stalk of herbaceous plant. (**c**,**d**) Vertical section of the ascoma. (**e**) Vertical section of the base of peridium. (**f**) Peridium. (**g**) Hamathecium. (**h**–**m**) Asci. (**n**–**s**) Ascospores. (**t**) Germinating ascospores. (**u**) Front and reverse of the colony on PDA. Scale bars: (**c**,**d**) = 100 μm, (**e**–**m**) = 30 μm, (**n**–**s**) = 10 μm, (**t**) = 20 μm.

**Table 1 jof-09-00612-t001:** Details of genes/loci with PCR primers and thermal cycling program for PCR amplification.

Genes/Loci	PCR Primers (Forward/Reverse)	PCR Thermal Cycle Protocols	References
ITS	ITS5/ITS4	^a^ Annealing at 55 °C for 15 s ^c^	[35]
LSU	LR0R/LR5		[36]
SSU	NS1/NS4	^a^ Annealing at 55 °C for 30 s ^c^	[35]
*tub*2	T1/Bt2b		[37]
*rpb*2	fRPB2-5F/fRPB2-7cR	^b^ Annealing at 57 °C for 50 s ^c^	[38]

Notes: ^a^ initial denaturation at 94 °C for 3 min, followed by 35 cycles of denaturation at 94 °C for 10s, elongation at 72 °C for 20 s; ^b^ initial denaturation at 95 °C for 3 min, followed by 35 cycles at 95 °C for 45 s, elongation at 72 °C for 1.5 min; ^c^ final extension at 72 °C for 10 min.

**Table 2 jof-09-00612-t002:** Names, Index Fungorum strain numbers, and corresponding GenBank accession numbers of the taxa used for phylogenetic analyses in this study.

Species Name	Strain Numbers	GenBank Accession Numbers				
		ITS	LSU	SSU	*tub*2	*rpb*2
** *Alloleptosphaeria clematidis* **	**MFLUCC 17-2071**	MT310604	MT214557	MT226674	NA	NA
** *All. iridicola* **	**CBS 143395**	MH107919	MH107965	NA	NA	NA
** *All. italica* **	**MFLUCC 14-0934**	KT454722	KT454714	NA	NA	NA
** *All. shangrilana* **	**HKAS 112210**	MW431059	MW431315	MW431058	NA	NA
** *Alternariaster bidentis* **	**CBS 134021**	KC609333	KC609341	NA	NA	KC609347
*Alt. bidentis*	CBS 134185	KC609334	KC609342	NA	NA	KC609348
** *Alt. centaureae-diffusae* **	**MFLUCC 14-0992**	KT454723	KT454715	KT454730	NA	NA
*Alt. centaureae-diffusae*	MFLUCC 15-0009	KT454724	KT454716	KT454731	NA	NA
*Alt. centaureae-diffusae*	HMJAU 60188	OL996125	OL897175	OL891810	OL898721	NA
*Alt. helianthi*	CBS 199.86	KC609336	KC609343	NA	NA	KC609349
** *Alt. helianthi* **	**CBS 327.69**	KC609335	KC584369	KC584627	NA	KC584494
*Alt. trigonosporus*	MFLU 15-2237	KY674857	KY674858	NA	NA	NA
** *Angularia xanthoceratis* **	**HMJAU 60197**	OM295683	OM295682	OM295681	OM304358	NA
** *Didymella exigua* **	**CBS 183.55**	EF192139	EU754155	EU754056	GU237525	EU874850
** *D. maydis* **	**CBS 588.69**	FJ427086	MH871149	EU754093	FJ427190	GU371782
*Heterosporicola beijingensis*	JZB3400001	MN733734	MN737597	MN733738	NA	NA
*H. beijingensis*	JZB3400002	MN733735	MN737598	MN733739	NA	NA
*H. beijingensis*	JZB3400003	MN733736	MN737599	MN733740	NA	NA
*H. beijingensis*	JZB3400004	MN733737	MN737600	MN733741	NA	NA
** *H. chenopodii* **	**CBS 448.68**	FJ427023	EU754187	EU754088	NA	NA
*H. chenopodii*	CBS 115.96	JF740227	EU754188	EU754089	NA	GU371775
** *H. dimorphospora* **	**CBS 345.78**	JF740203	GU238069	GU238213	NA	NA
*H. dimorphospora*	CBS 165.78	JF740204	JF740281	JF740098	NA	NA
*Leptosphaeria chatkalica*	YGS22	MW886101	MW886099	MW886100	NA	NA
** *L. cichorii* **	**MFLUCC 14-1063**	KT454720	KT454712	KT454728	NA	NA
** *L. cirsii* **	**MFLUCC 14-1170**	NR155328	NG059725	NA	NA	NA
*L. conoidea*	CBS 125977	JF740202	JF740280	NA	NA	NA
*L. conoidea*	CBS 616.75	JF740201	JF740279	JF740099	KT389804	KT389639
*L. conoidea*	FeF93	MZ492958	NA	NA	NA	NA
*L. doliolum*	MFLUCC 15-1875	KT454727	KT454719	KT454734	NA	NA
*L. doliolum*	CBS 155.94	JF740207	JF740282	NA	JF740146	NA
** *L. doliolum* **	**CBS 505.75**	JF740205	GQ387576	GQ387515	JF740144	KY064035
*L. doliolum*	CBS 541.66	JF740206	JF740284	NA	JF740145	NA
*L. doliolum*	CBS 130000	JF740210	NA	NA	JF740149	NA
** *L. ebuli* **	**MFLUCC 14-0828**	KP744446	KP744488	KP753954	NA	NA
*L. errabunda*	CBS 617.75	JF740216	JF740289	NA	JF740150	NA
*L. errabunda*	CBS 125978	JF740217	JF740290	NA	JF740151	NA
*L. errabunda*	CBS 129998	JF740219	MH877027	NA	JF740153	NA
*L. italica*	MFLU15-0174	NA	KT783670	NA	NA	NA
*L. irregularis*	MFLUCC 15-1118	KX856056	KX856055	NA	NA	NA
*L. macrocapsa*	CBS 640.93	JF740237	JF740304	NA	JF740156	NA
*L. pedicularis*	CBS 390.80	JF740224	JF740294	NA	JF740155	NA
*L. pedicularis*	CBS 126582	JF740223	JF740293	NA	NA	NA
*L. proteicola*	CPC:18289	JQ044439	JQ044458	NA	NA	NA
** *L. regiae* **	**MFLUCC 18-1137**	MN244201	MN244171	MN244177	NA	NA
*L. sclerotioides*	CBS 144.84	JF740192	JF740269	NA	NA	NA
*L. sclerotioides*	CBS 148.84	JF740193	JF740270	NA	NA	NA
*L. sclerotioides*	FeF422	MZ492959	NA	NA	NA	NA
*L. sclerotioides*	P10	MT996500	MT996501	NA	MT996502	MT996503
*L. sclerotioides*	P9	MT996499	MT992704	NA	MT989358	MT992705
*Leptosphaeria* sp.	LW119	MH128282	NA	NA	NA	NA
*Leptosphaeria* sp.	LW113	MH128276	NA	NA	NA	NA
*L. slovacica*	CBS 389.80	JF740247	JF740315	JF740101	NA	NA
*L. slovacica*	CBS 125975	JF740248	JF740316	NA	NA	NA
*L. sydowii*	CBS 385.80	JF740244	JF740313	NA	JF740157	NA
*L. sydowii*	CBS 125976	JF740245	JF740314	NA	JF740158	NA
** *L. urticae* **	**MFLU 18-0591**	MK123333	MK123332	MK123329	NA	NA
*L. urticae*	FeF166	MZ492960	NA	NA	NA	NA
*L. veronicae*	CBS 126583	JF740255	JF740321	NA	JF740161	NA
*L. veronicae*	CBS 145.84	JF740254	JF740320	NA	JF740160	NA
** *L. yunnanensis* **	**CGMCC 3.23748**	OP494319	OP494327	OP494333	OP476696	NA
*L. yunnanensis*	CGMCC 3.23749	OP494320	OP494328	OP494334	OP476697	NA
*L. yunnanensis*	HKAS 124671	OP494321	OP494329	OP494335	OP476698	NA
** *L. zhaotongensis* **	**HKAS 124664**	OP494318	OP494326	OP494332	OP476695	NA
*L. zhaotongensis*	HMAS 352282	OQ446062	OQ446132	OQ448836	OQ511597	
** *Neoleptosphaeria jonesii* **	**MFLUCC 16-1442**	KY211869	KY211870	KY211871	NA	NA
** *N. rubefaciens* **	**CBS 223.77**	JF740243	JF740312	NA	NA	NA
*N. rubefaciens*	CBS 387.80	JF740242	JF740311	NA	NA	NA
** *Ochraceocephala foeniculi* **	**CBS 145654**	MN516753	MN516774	MN516743	MN520147	MN520145
*O. foeniculi*	Di3AF15	MN516766	MN516783	MN516752	NA	NA
*Paraleptosphaeria dryadis*	CBS 643.86	JF740213	GU301828	KC584632	NA	GU371733
*Pa. dryadis*	CBS 743.86	AF439461	NA	NA	NA	NA
** *Pa. kunmingensis* **	**KUNCC 23-12732**	OP494316	OP494324	OP494330	OP476693	OP476691
*Pa. kunmingensis*	KUNCC 23-12731	OQ446060	OQ446130	OQ448834	OQ511598	OQ455053
*Pa. macrospora*	CBS 114198	JF740238	JF740305	NA	NA	NA
** *Pa. nitschkei* **	**CBS 306.51**	JF740239	JF740308	NA	KT389833	KT389660
*Pa. nitschkei*	MFLUCC 13-0688	KR025860	KR025864	NA	NA	NA
** *Pa. orobanches* **	**CBS 101638**	JF740230	JF740299	A	NA	NA
*Pa. padi*	MFLU 15-2756	KY554203	KY554198	KY554201	NA	NA
*Pa. polylepidis*	APA-2999	MK795714	MK795717	NA	NA	NA
*Pa. praetermissa*	CBS 114591	JF740241	JF740310	NA	NA	NA
** *Pa. rubi* **	**MFLUCC 14-0211**	KT454726	KT454718	KT454733	NA	NA
** *Pa. rumicis* **	**CBS 522.78**	KF251144	KF251648	NA	NA	NA
*Plenodomus agnitus*	CBS 121.89	JF740194	JF740271	NA	KY064053	KY064036
*Pl. agnitus*	CBS 126584	JF740195	JF740272	NA	NA	NA
*Pl. agnitus*	MFLU 15-0039	KP744459	KP744504	NA	NA	NA
*Pl. artemisiae*	KUMCC 18-0151	MK387920	MK387958	MK387928	NA	MK435607
** *Pl. artemisiae* **	**KUMCC 20-0200A**	MT957062	MT957055	MT957048	NA	NA
*Pl. artemisiae*	KUMCC 20-0200B	MT957063	MT957056	MT957049	NA	NA
*Pl. biglobosus*	CBS 119951	JF740198	JF740274	JF740102	KY064054	KY064037
*Pl. biglobosus*	CBS 127249	JF740199	JF740275	NA	NA	NA
** *Pl. changchunensis* **	**CCMJ5011**	OL996123	OL897174	OL984031	NA	NA
*Pl. changchunensis*	CCMJ5012	OL996124	OL966928	OL984032	OL898716	OL944508
** *Pl. collinsoniae* **	**VT02**	MN653010	MN982862	MN652269	NA	NA
*Pl. collinsoniae*	CBS 120227	JF740200	JF740276	NA	KY064056	KY064039
*Pl. collinsoniae*	KNU-AP100C	LC550566	LC550568	NA	NA	NA
*Pl. collinsoniae*	KNU-20-A1	LC591836	NA	NA	LC591846	LC591841
*Pl. collinsoniae*	KNU-20-A2	LC591837	NA	NA	LC591847	LC591842
*Pl. collinsoniae*	KNU-20-A3	LC591838	NA	NA	LC591848	LC591843
*Pl. collinsoniae*	KNU-20-A4	LC591839	NA	NA	LC591849	LC591844
*Pl. collinsoniae*	KNU-20-C4	LC591840	NA	NA	LC591850	LC591845
*Pl. collinsoniae*	FeC109	MW446975	NA	NA	NA	NA
*Pl. collinsoniae*	EF194	MK842112	NA	NA	NA	NA
*Pl. confertus*	CBS 375.64	AF439459	JF740277	NA	KY064057	KY064040
** *Pl. congestus* **	**CBS 244.64**	AF439460	JF740278	NA	KY064058	KY064041
** *Pl. deqinensis* **	**CGMCC 3.18221**	KY064027	KY064031	NA	KY064052	KY064034
** *Pl. dezfulensis* **	**IRAN 4159C**	MZ048609	NA	NA	MZ043102	MZ043104
*Pl. dezfulensis*	SCUA-Ahm-S41-2	MZ048610	NA	NA	MZ043103	MZ043105
** *Pl. enteroleucus* **	**CBS 142.84**	JF740214	JF740287	NA	KT266266	KY064042
*Pl. enteroleucus*	CBS 831.84	JF740215	JF740288	NA	KT266270	NA
*Pl. enteroleucus*	F-146,176	MN910295	MN910294	NA	NA	NA
*Pl. enteroleucus*	ICMP:10937	KT309810	KT309635	NA	KT309399	NA
*Pl. fallaciosus*	CBS 414.62	JF740222	JF740292	NA	NA	KY064043
** *Pl. guttulatus* **	**MFLUCC 15-1876**	KT454721	KT454713	KT454729	NA	NA
*Pl. hendersoniae*	CBS 113702	JF740225	JF740295	NA	KT266271	KY064044
*Pl. hendersoniae*	CBS 139.78	JF740226	JF740296	NA	NA	NA
*Pl. hendersoniae*	LTO	MF795790	NA	NA	NA	MF795832
** *Pl. influorescens* **	**CBS 143.84**	JF740228	JF740297	NA	KT266267	KY064045
*Pl. influorescens*	PD 73/1382	JF740229	JF740298	NA	KT266273	NA
*Pl. libanotidis*	CBS 113795	JF740231	JF740300	NA	KY064059	KY064046
** *Pl. lijiangensis* **	**KUMCC 18-0186**	MK387921	MK387959	MK387929	NA	NA
*Pl. lindquistii*	CBS 386.80	JF740232	JF740301	NA	NA	NA
*Pl. lindquistii*	CBS 381.67	JF740233	JF740302	NA	NA	NA
*Pl. lindquistii*	MF-Ha 16-005	MK495988	NA	NA	MK501790	NA
*Pl. lingam*	AFTOL-ID 277	KT225526	DQ470946	DQ470993	NA	DQ470894
*Pl. lingam*	CBS 275.63	JF740234	JF740306	JF740103	KT389841	KT389669
** *Pl. lingam* **	**CBS 260.94**	JF740235	JF740307	NA	MZ073915	KY064047
*Pl. lingam*	CBS 147.24	MH854784	MH866288	NA	MZ073914	NA
** *Pl. lupini* **	**CBS 248.92**	JF740236	JF740303	NA	KY064061	KY064048
** *Pl. pimpinellae* **	**CBS 101637**	JF740240	JF740309	NA	KY064062	NA
** *Pl. salviae* **	**MFLUCC 13-0219**	KT454725	KT454717	KT454732	NA	NA
*Pl. sinensis*	KUMCC 18-0152	MK387923	MK387961	MK387931	NA	NA
*Pl. sinensis*	KUMCC 18-0153	MK387922	MK387960	MK387930	NA	MK435608
*Pl. sinensis*	KUN-HKAS 102227	MK387924	MK387962	MK387932	NA	NA
*Pl. sinensis*	MFLUCC 17-0757	MF072722	MF072718	MF072720	NA	NA
** *Pl. sinensis* **	**MFLUCC 17-0767**	MF072721	MF072717	MF072719	NA	NA
*Pl. sinensis*	KNU-GW1901	LC550567	LC550569	LC550570	NA	NA
*Pl. sinensis*	KUMCC 20-0204	MT957064	MT957057	MT957050	NA	NA
*Pl. tracheiphilus*	CBS 551.93	JF740249	JF740317	JF740104	MZ073918	KY064049
** *Pl. tracheiphilus* **	**CBS 127250**	JF740250	JF740318	NA	MZ073919	NA
*Pl. tracheiphilus*	MUCL 38481	MW810293	MW715037	NA	MZ073920	NA
*Pl. tracheiphilus*	ATCC 26007	MZ049614	MW959165	NA	MZ073908	MZ073893
*Pl. tracheiphilus*	IS3-15	MK461058	NA	NA	NA	NA
** *Pl. triseptatus* **	**MFLUCC 17-1345**	MN648452	MN648451	MN648453	NA	NA
** *Pl. visci* **	**CBS 122783**	JF740256	EU754195	EU754096	KY064063	KY064050
*Pl. visci*	CPC:35314	MT223830	MT223922	NA	NA	MT223696
*Pl. visci*	CPC:35315	MT223831	MT223923	NA	NA	NA
*Pl. visci*	CPC:35316	MT223832	MT223924	NA	NA	NA
*Pl. wasabiae*	CBS 120119	JF740257	JF740323	NA	KT266272	NA
*Pl. wasabiae*	CBS 120120	JF740258	JF740324	NA	NA	NA
** *Pl. zhaotongensis* **	**CGMCC 3.23746**	OP494317	OP494325	OP494331	OP476694	OP476692
*Pl. zhaotongensis*	CGMCC 3.23747	OQ446061	OQ446131	OQ448835	OQ511599	OQ455054
** *Praeclarispora artemisiae* **	**KUMCC 20-0201A**	MT957060	MT957053	MT957046	NA	NA
*Pr. artemisiae*	KUMCC 20-0201B	MT957061	MT957054	MT957047	NA	NA
** *Pseudoleptosphaeria etheridgei* **	**CBS 125980**	JF740221	JF740291	NA	NA	MT394686
*Querciphoma carteri*	CBS 101633	KF251210	GQ387593	GQ387532	KF252701	NA
*Querciphoma carteri*	CBS 105.91	KF251209	GQ387594	GQ387533	KF252700	NA
** *Sclerenchymomyces clematidis* **	**MFLUCC 17-2180**	MT310605	MT214558	MT226675	NA	NA
*Sphaerellopsis artemisiae*	KUMCC 20-0202A	MT957065	MT957058	MT957051	NA	NA
*Sp. artemisiae*	KUMCC 20-0202B	MT957066	MT957059	MT957052	NA	NA
*Sp. filum*	CBS 234.51	KP170655	KP170723	NA	KP170704	NA
*Sp. filum*	CBS 235.51	KP170656	KP170724	NA	KP170705	NA
** *Sp. filum* **	**CBS 317.68**	KP170657	KP170725	NA	KP170706	NA
** *Sp. hakeae* **	**CPC:29566**	KY173466	KY173555	NA	NA	NA
** *Sp. isthmospora* **	**HKAS 102225A**	MK387925	MK387963	NA	NA	NA
*Sp. isthmospora*	HKAS 102225B	MK387926	MK387964	MK387934	NA	NA
*Sp. macroconidialis*	CBS 233.51	KP170658	KP170726	NA	KP170707	NA
** *Sp. macroconidialis* **	**CBS 658.78**	KP170659	KP170727	NA	KP170708	NA
*Sp. macroconidialis*	CPC:21113	KP170660	KP170728	NA	KP170709	NA
** *Sp. paraphysata* **	**CPC:21841**	KP170662	KP170729	NA	KP170710	NA
*Sp. paraphysata*	KUMCC 18-0195	MK387927	MK387965	MK387935	NA	NA
*Subplenodomus apiicola*	CBS 285.72	JF740196	GU238040	GU238211	NA	NA
*Su. apiicola*	CBS 421.50	MH856699	MH868215	NA	NA	NA
*Su. drobnjacensis*	CBS 270.92	JF740212	JF740286	NA	NA	NA
*Su. drobnjacensis*	CBS 269.92	JF740211	JF740285	JF740100	NA	NA
** *Su. galicola* **	**MFLU 15-1368**	KY554204	KY554199	NA	NA	NA
*Su. valerianae*	CBS 630.68	JF740251	GU238150	GU238229	NA	NA
*Su. valerianae*	CBS 499.91	JF740252	JF740319	NA	NA	NA
*Su. violicola*	CBS 306.68	FJ427083	GU238156	GU238231	KT389849	NA
*Shiraia bambusicola*	GZAAS2.0703	GQ845412	KC460981	NA	NA	NA
*Shiraia bambusicola*	GZAAS2.0629	GQ845415	KC460980	NA	NA	NA
*Tzeanania taiwanensis*	NTUCC 17-005	MH461123	MH461120	MH461126	MH461132	NA
*Tzeanania taiwanensis*	NTUCC 17-006	MH461124	MH461121	MH461127	MH461133	NA

Notes: The ex-types are in bold, and newly generated sequences are shown in blue. NA: sequence data are not available.

## Data Availability

Not applicable.

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
