# Peer review of "Morphological and Phylogenetic Characterisations Reveal Four New Species in Leptosphaeriaceae (Pleosporales, Dothideomycetes)"

_jof, 2023, doi:10.3390/jof9060612_

Round 1

Reviewer 1 Report

I enjoyed reading this manuscript. The paper is very well written. The description of the new species, the photographs, and the molecular analysis are all high quality.

I have one main concern though. For each new species, the authors provided a brief discussion on how the new species are anatomically different to the closest sequenced species in their phylogeny. This is great, however, as the authors mentioned, most species in Leptosphaeriaceae are not sequenced yet. The authors need to provide a brief discussion on how each new species differs from similar species regardless of the availability of sequences for those similar species. Otherwise, it is not clear if we are dealing with new species or just with already known species that have not been sequenced yet. 

Other minor comments are the following:

In Table 2, page 5, all three instances of L. yunnanensis are written in bold. Only the first name should be in bold.

In page 11, line 173, the authors use “,thickness,” as part of the description. Something is missing there, or the last coma might be unnecessary?

In page 11, line 179 change “1.4µm” for “1.4 µm”

In page 12, lines 218–219 the sentence “Ostiole apex with a conical with papilla.” Sounds wrong, perhaps “Ositole apex conical and with papilla” or “Ositole apex conical, with papilla”

Throughout the manuscript the authors use the term “Hermathecium”, I could not find a definition for that term… perhaps is a misspelling of hamathecium?

Author Response

Reviewer Comments and Responses

Responses to Reviewer 1

Authors are obliged to you for your positive and encouraging comments.

Comment:

I enjoyed reading this manuscript. The paper is very well written. The description of the new species, the photographs, and the molecular analysis are all high quality.

I have one main concern though. For each new species, the authors provided a brief discussion on how the new species are anatomically different to the closest sequenced species in their phylogeny. This is great, however, as the authors mentioned, most species in Leptosphaeriaceae are not sequenced yet. The authors need to provide a brief discussion on how each new species differs from similar species regardless of the availability of sequences for those similar species. Otherwise, it is not clear if we are dealing with new species or just with already known species that have not been sequenced yet.

Response: Thank you very much for your valuable comments. We compared each of our new species with known species with/without molecular data as below and added in the note and discussion section;

  • Leptosphaeria yunnanensis was reported as an asexual form and compared to Leptosphaeria cichorii in the note section, page 11, lines 204–208.
  • Leptosphaeria zhaotongensis: We compared with known species that have no molecular data in Shoemaker (1985) and added in the discussion section, page 19–20, lines 400–414.
  • Paraleptosphaeria kunmingensis: All species of Paraleptosphaeria have sequence data in this study (Table 2), page 5.
  • Plenodomus zhaotongensis: We compared with known species and added in the note section, page 17, lines 341–345.

Comment: In Table 2, page 5, all three instances of L. yunnanensis are written in bold. Only the first name should be in bold.

Response: Accepted and revised, in Table 2, page 5.

Comment: In page 11, line 173, the authors use “, thickness,” as part of the description. Something is missing there, or the last coma might be unnecessary?

Response: Accepted and revised, page 10, lines 176.

Comment: In page 11, line 179 change “1.4µm” for “1.4 µm”

Response: Accepted and revised, page 10, lines 182.

Comment: In page 12, lines 218–219 the sentence “Ostiole apex with a conical with papilla.” Sounds wrong, perhaps “Ositole apex conical and with papilla” or “Ositole apex conical, with papilla”

Response: Accepted and revised to Ostiole apex conical and with papilla, page 12, lines 227–228.

Comment: Throughout the manuscript the authors use the term “Hermathecium”, I could not find a definition for that term… perhaps is a misspelling of hamathecium?

Response: Sorry, it’s a misspelling, we changed it to “Hamathecium”, lines 263, lines 295, lines 299, lines 349.

Reviewer 2 Report

The paper described four new species of fungi isolated from a grassland ecosystem. It is well written and the experiments support the identification of these new species. I have very minimal comments on the manuscript. 

Author Response

Responses to Reviewer 2

Comment:

The paper described four new species of fungi isolated from a grassland ecosystem. It is well written and the experiments support the identification of these new species. I have very minimal comments on the manuscript.

Response: Authors are obliged to you for your positive and encouraging comments.

Comment: perhaps you can also mention some economic importance, if any, of this group.

Response: Accepted and revised. “In the order Pleosporales, the family Leptosphaeriaceae contains economically significant plant pathogens [2] i.e., Zhang et al. [8] reported Leptosphaeria species cause a serious disease of oilseed rape (Brassica napus, Canola) in China.” page 1, lines 34–37.

Comment: kindly include GPS data

Response: Thank you very much for your comment. We agree, that these data are important for the readers. However, we have provided these data in the taxonomy section under the “Material examined”. If we add them again, it will repeat the same data.

Comment: Line 164; Line 210; Line 244; Line 293 italicized

Response: Accepted and revised. Line 167; Line 219; Line 253; Line 302.

Reviewer 3 Report

The authors make an important contribution to the leptophaeriaceae family. They carried out a polyphasic study, in which they perfectly portrayed the fungal structures. I can only recommend a couple of changes to the text. 

Line 30: the name of the researcher who introduced the family is not available 

Figure 1: the font size is  too small 

Line 164: italic 

Line 179: guttulate conidia are observed in figure 2, verify if this is the case and add in the description. 

Line 210: italic 

Line 244: italic 

Line 286: What is hermathecium? 

Line 290: What is hermathecium? 

Line 293: italic 

Line 340: What is hermathecium?

Author Response

Responses to Reviewer 3

Comment:

The authors make an important contribution to the leptophaeriaceae family. They carried out a polyphasic study, in which they perfectly portrayed the fungal structures. I can only recommend a couple of changes to the text.

Response: Authors are obliged to you for your positive and encouraging comments.

Comment: Line 30: the name of the researcher who introduced the family is not available

Response: Accepted and revised, page 1, lines 31.

Comment: Figure 1: the font size is too small

Response: Accepted and revised, page 8,9.

Comment: Line 164; Line 210; Line 244; Line 293: italic

Response: Accepted and revised. Line 167; Line 219; Line 253; Line 302.

Comment: Line 179: guttulate conidia are observed in figure 2, verify if this is the case and add in the description.

Response: Accepted and revised. page 10, lines 183.

Comment: Line 286, Line 290, Line 340, What is hermathecium?

Response: Sorry, this is a misspelling so we revised it as, “Hermathecium was changed to Hamathecium” lines 263, lines 295, lines 299, lines 349.
